# Saliva Proteomics as Fluid Signature of Inflammatory and Immune-Mediated Skin Diseases

**DOI:** 10.3390/ijms22137018

**Published:** 2021-06-29

**Authors:** Anna Campanati, Emanuela Martina, Federico Diotallevi, Giulia Radi, Andrea Marani, Davide Sartini, Monica Emanuelli, George Kontochristopoulos, Dimitris Rigopoulos, Stamatis Gregoriou, Annamaria Offidani

**Affiliations:** 1Dermatological Unit, Department of Clinical and Molecular Sciences, Polytechnic Marche University, 60100 Ancona, Italy; ema.martina@gmail.com (E.M.); federico.diotallevi@gmail.com (F.D.); radigiu1@gmail.com (G.R.); andreamarani.med@yahoo.com (A.M.); annamaria.offidani@ospedaliriuniti.marche.it (A.O.); 2Biochemistry, Department of Clinical Sciences, Polytechnic Marche University, 60100 Ancona, Italy; d.sartini@univpm.it (D.S.); m.emanuelli@univpm.it (M.E.); 3Department of Dermatology-Venereology, Faculty of Medicine, National and Kapodistrian University of Athens, Andreas Sygros Hospital, 124 62 Athens, Greece; gkontochris@gmail.com (G.K.); dimitrisrigopoulos54@gmail.com (D.R.); stamgreg@yahoo.gr (S.G.)

**Keywords:** saliva, psoriasis, oral lichen plants, vitiligo, atopic dermatitis, blistering diseases

## Abstract

Saliva is easy to access, non-invasive and a useful source of information useful for the diagnosis of serval inflammatory and immune-mediated diseases. Following the advent of genomic technologies and -omic research, studies based on saliva testing have rapidly increased and human salivary proteome has been partially characterized. As a proteomic protocol to analyze the whole saliva proteome is not currently available, the most common aim of the proteomic analysis is to discriminate between physiological and pathological conditions. The salivary proteome has been initially investigated in several diseases: oral squamous cell carcinoma and oral leukoplakia, chronic graft-versus-host disease, and Sjögren’s syndrome. Otherwise, salivary proteomics studies in the dermatological field are still in the initial phase, thus the aim of this review is to collect the best research evidence on the role of saliva proteomics analysis in immune-mediated skin diseases to understand the direction of research in this field. The results of PRISMA analysis reported herein suggest that human saliva analysis could provide significant data for the diagnosis and prognosis of several immune-mediated and inflammatory skin diseases in the next future.

## 1. Introduction

Salivary proteome analysis has progressively evolved in different biomedical fields of research such as genetics, molecular biology, medicine, and dentistry [1,2] in the last decade [3,4].

Several data from the literature have been reported on the opportunity to recur to saliva firstly as a diagnostic fluid to detect oral diseases such as periodontitis [5], oral squamous cell carcinoma [1,2], burning mouth syndrome [6] and Sjögren’s syndrome [7]. Moreover, the saliva proteome has already been used to evaluate its change in immune-mediated inflammatory systemic diseases, e.g., diabetes mellitus [8], cystic fibrosis [9], Parkinson disease [10], and multiple sclerosis [11], revealing the great potential of proteomics in biomarker identification and providing new insight into the molecular mechanisms underlying several systemic diseases.

Reasons for increasing interest in saliva as an attractive body fluid for diagnosis of different systemic diseases are numerous: the collection of saliva samples is usually easy to perform, economical, and safe, and moreover, it can be considered a non-invasive and well-tolerated procedure by patients because its collection is not painful [12].

The most common aim of the proteomic analysis is to discriminate between physiological and pathological conditions, in this view, the purpose of this PRISMA review is to briefly describe the most salient aspects of current proteomic researches carried out on human saliva in inflammatory and immune-mediated skin diseases, with particular regard to its potential use as a diagnostic fluid.

In 2008, the term “salivaomics” was developed to highlight the many “omics” contained in saliva, including the genome, transcriptome, proteome, metabolome, and microbiome [13]. Salivaomics has been extensively explored in recent years as more powerful analytical techniques have become available. In saliva, over 70% of the DNA is human, with the remaining 30% belonging to the oral microbiota [14].

Saliva samples can be tested using polymerase chain reaction (PCR) and sequencing arrays. Salivary DNA analysis is used to examine for abnormal DNA methylation, which is the earliest epigenetic sign of neoplastic changes [15]. The word “proteome” refers to all proteins found in the mouth. Saliva comprises around 2000 proteins with a wide range of biological functions [16], with about a quarter of salivary proteins detectable in plasma. NMR spectroscopy, as well as gas and liquid chromatography–mass spectrometry, are used in proteomics (GC-MS and LC-MS).

Two-dimensional polyacrylamide gel electrophoresis (2D-PAGE) and capillary electrophoresis with electrochem are used in this line of study.

2D-PAGE, which preceded 2D-DIGE, fractionates proteins based on their isoelectric points in one dimension and apparent molecular weight in the other Kondo, T. Cancer biomarker development and two-dimensional difference gel electrophoresis (2D-DIGE) [17]. The enzyme-linked immunosorbent test (ELISA) has been utilized for several purposes, including diagnostics and quality control [18].

The simplest format is direct ELISA, which requires an antigen and an antigen-specific enzyme-conjugated antibody [19].

Many variables can affect the composition and overall volume of spit. The time of day, hydration, body posture, drug intake, smoking, psychological stimulation, dietary assumption, and other systemic circumstances can all influence saliva characteristics in a single participant [3].

In clinical studies, saliva is typically collected at rest (“unstimulated saliva”) after at least 1 h of fasting, without drinking or smoking; the patient must be comfortably seated, prevent oro-facial movements for 5 min, and rinse their mouth with deionized water right before sampling [20].

The gold standard way is to use specific instruments to drain saliva [21]. In the literature, there are disagreements on centrifugation and speed, the use of PIC (protease inhibitor cocktail), and storage temperature. The majority of writers suggested using a protease inhibitor combination to stabilize the substrate; also, the samples obtained must be promptly kept in ice containers and, following processing, held at 80 °C [22]. All of these stages are required for the suppression of bacterial growth and the minimal damage of salivary proteins.

## 2. Materials and Methods

This systematic review was based on the approach developed by Arksey and O’Malley [23], which involves five key steps: identify the research question; identify relevant studies; study selection; chart the data; and, collate, summarize, and report the results. The Preferred Reporting Items for Systematic Reviews and Meta-Analysis (PRISMA) extension for scoping reviews criteria to guide the conduct and reporting of the review was used [24,25].

### 2.1. Identify the Research Question

To identify the research question, a brainstorming approach, involving the entire research group, was used. The research team consisted of six healthcare professionals experienced in inflammatory and immune-mediated skin disease and proteomic: six Ph.D. doctors and four clinicians.

During the first meeting, the research team identified the research question and established the research strategy. The research question was: “is saliva proteomics useful for diagnosis, prognosis, response to treatment evaluation in immune mediates skin diseases?”.

### 2.2. Identify Relevant Studies

The keywords used were “saliva AND proteomics”, “saliva AND immune-mediated skin diseases”, “saliva AND psoriasis”, “saliva AND suppurative hidradenitis”, “saliva AND blistering skin diseases”, “saliva AND atopic dermatitis”, “saliva AND oral lichen planus” “saliva AND vitiligo”. In this first phase, 139 records were identified. The removed records after duplicates totaled 128.

Inclusion criteria: studies reporting on saliva proteomics in IMID, studies published in the English language, abstract available, no restriction on design of the study was considered, and randomized controlled trial, case-control study, cross-sectional study, case reports and series were included. Exclusion criteria: studies reporting on review articles.

### 2.3. Study Selection

The selection of the relevant studies took place in three steps. In the first step, four researchers (A.C., E.M., D.M., and M.E.) independently made a selection of the articles based on the title. Any disagreement was solved by consulting a senior investigator (AO). The second step consisted of evaluating the abstracts. At least two members of the research team (F.D. and G.R.) independently assessed each abstract. The research team resolved all discrepancies through consensus. Eighty-four papers were excluded and 44 were have been assessed for full-text analysis. The third phase consisted of the critical evaluation of the full text of the selected articles. A final sample of 37 studies was included in the qualitative synthesis.

### 2.4. Data Extraction

A data extraction form was first designed by A.C. prior to the extraction of the data, to expedite the process.

In order to answer the research question, the following information was extracted from the included articles: Name of the author(s) and date of publication; study design; study population; sample size; saliva proteomics investigated; type of IMID; measured outcomes; findings of the study and the recommendations from the study.

## 3. Results

The PRISMA study flowchart is shown in Figure 1. Our search identified 128 records after removing duplicates. After scanning the titles and abstracts, 84 citations were excluded, and 44 were have been assessed for full-text eligibility. After examining the full text, 37 case-control, case series studies, randomized controlled trials and meta-analyses studies were considered eligible and included in this study.

## 4. Discussion

Immune-mediated and inflammatory skin diseases, for which consistent salivary proteomics analysis studies have been produced, can be summarized as follows: Psoriasis Atopic dermatitis Blistering diseases Lichen planus Vitiligo

### 4.1. Psoriasis

Psoriasis is a chronic immune-mediated inflammatory skin condition, currently interpreted as a multisystemic disorder. Several pieces of evidence have linked psoriasis to comorbidities, such as cardiovascular and oral ones, and suggested the need for reliable biomarkers, capable of identifying a pathological status and/or a therapeutic response [26,27,28,29,30]. In the past years, biomarkers research in psoriasis, focused on blood and skin samples, genetics or transcriptomics, led to contrasting results. Thus, scientific efforts have been directed to an easily manageable, fascinating biological fluid: saliva [31].

Salivary “proteome” consists of more than 2000 proteins contained in saliva. Of this amount, only a fraction has been investigated in psoriatic patients, since studies conducted so far focused only on few, specific biomarkers. However, we have analyzed this limited but emerged part, and assumed future perspectives for this promising field.

Salivary levels of acute-phase proteins such as alpha amylase (sAA), haptoglobin and C-reactive protein (CRP) were assessed in several studies. Psoriatic patients had a statistically significant increase of CRP, associated with the inflammatory nature of the disease, and, as known, the prognostic significance CRP has for the course of psoriasis. Analogously, increased salivary levels of haptoglobin were found, suggesting a local mechanism against psoriasis. Soudan and colleagues revealed a higher concentration rate of sAA in 20 psoriatic patients compared to the controls; these salivary changes were not related to the severity and the duration of the disease [31,32].

For a more in-depth exploration of the altered immune role of saliva in course of psoriasis, salivary IgA levels were evaluated. Findings were contradictory, probably because of the different type of investigated disease: there was no statistically significant difference between psoriatic patients and healthy controls, but patients with PASI > 10 had lower IgA levels compared with those of patients with PASI < 10, suggesting that patients affected by moderate-to-severe psoriasis might be at high risk of developing microbial infections [27]. A Danish study revealed lower salivary levels of NGAL (neutrophil gelatinase-associated lipocalin) and transferrin in psoriatic patients compared with patients with periodontitis and orally healthy controls [33].

Several cytokines were assayed in psoriatic patients’ saliva. Ganzetti and colleagues demonstrated that psoriatic patients had higher salivary IL-1β levels than controls, and TNF-α inhibitors treatment significantly reduced IL-1β levels compared with baseline, but without reaching the normal value [34,35]. The same author investigated the expression of the two other primary cytokines, IL-6 and TNF-α, and other cytokines, such as transforming growth factor (TGF)-β1, IL-8, interferon (IFN)-γ, IL-17A, IL-4, IL-10, monocyte chemoattractant protein (MCP)-1, microphage inflammatory protein (MIP)-1a, and MIP-1b. Psoriatic patients had significantly higher salivary IL1β, TNF-α, TGF-β, and MCP-1 levels than controls [36,37]. Skutnik-Radziszewska and colleagues showed that also IL-2 level considerably increased in the saliva of psoriatic, whereas IL-10 content decreased, indicating a probable imbalance between Th1 and Th2 cells in salivary glands [38].

An interesting study, instead, explored saliva proteomics in psoriasis from a “metabolic point of view”. Authors revealed that concentration of peroxidase, catalase and superoxide dismutase was significantly higher in unstimulated saliva, and that of CAT and SOD significantly higher in stimulated saliva, of psoriatic patients compared with healthy subjects. This suggested a redox imbalance in course of psoriasis, with the prevalence of oxidation reactions [39]. Further knowledge on salivary proteome may be obtained by exploring another relevant metabolism, the one of vitamin D, often lacking in psoriatic patients’ serum, as recently done in a study regarding Recurrent Aphthous Stomatitis [40].

Some aspects of the “salivary signature” of psoriasis, such as increased dosage of certain cytokines or acute-phase proteins, are definitively emerging, so that saliva already now could be a valid non-invasive tool for monitoring inflammation. However, data need to be confirmed and expanded by further studies with larger sample size and wider salivary profiling, and concerning salivary changes in different forms of psoriasis, such as arthropathic or palm-plantar, of which a study has already highlighted the difference in the secondary structure of the proteins compared to the vulgar form, and after treatment [41]. Proteomic technologies seem to be promising and could be applied to saliva to achieve these goals [42]. We would encourage researchers to examine the diagnostic and prognostic role of IL1-β, TNF-α, IL-2 and IFN-γ in different psoriasis clinical forms.

### 4.2. Oral Lichen Planus

Oral lichen planus (OLP) is a specific chronic inflammatory disease of the oral mucosa, with an incidence of 0.5 percent to 2% in adults and a minor female predominance [43]. Different features can be identified clinically: reticular (the most common type), erythematous, ulcerative or erosive, plaque-like, bullous, or popular [44,45]. The histopathology of OLP is typical, with a conspicuous lymphocyte infiltrate at the epithelial interface, acanthosis, and basal cell layer degeneration [46]. The deposit of Immunoglobulin M as colloid bodies and C3 in granular and linear patterns in the basement membrane layer can be observed using direct immunofluorescence (DIF) [47]. While the specific causes of OLP remain unclear, autoreactive T cells are thought to play a key role. Stress, HCV and viral diseases, and medications have all been identified as potential risk/trigger factors [48]. For its chance of malignant transformation (0.04–1.74 per year) in squamous cell carcinoma, OLP has been identified as a premalignant lesion (OSCC) [49]. Patients with OLP experience burning and itching symptoms that progress to extreme pain in the erosive form of the disease; the disease has a significant negative effect on the quality of life leading to impairments of everyday tasks such as feeding and oral hygiene [50]. Published articles on salivary biomarkers in OLP are relatively new, and they concern OLP diagnosis, especially early detection of malignant transformation. Talungchit et al. used a proteomic technique on saliva with two-dimensional gel electrophoresis accompanied by mass spectrometry and recruit five OLP patients and five stable controls in 2018. The investigators used an ELISA test to identify three proteins that could play a role in OLP patients (cystatin SA, chain C of human complement component C3c, and chain B of fibrinogen fragment D) [51]. Cystatin SA belongs to the cystatin superfamily, a group of cysteine protease inhibitors with antimicrobial activity, while fibrinogen fragments D and C3c play a key role in inflammation. In reality, fibrinogen expression and C3 deposition are common in OLPs treated with IFD [47]. Another research conducted in 2017 [52] focused on a new and more complex panel of proteins. The researchers looked at 108 proteins that were differentially expressed in OLP patients compared to healthy controls using mass spectrometry. The absence of proteins required for lubrication and viscoelasticity was the first discovery, confirming the xerostomia symptom commonly mentioned by patients. The authors discussed the recognized roles of each peptide and attempted to correlate protein expression in saliva with histological findings in OLP. S100A8 and S100A9 (also known as MRP8 and MRP14) are calcium- and zinc-binding proteins that play a role in IL-17-mediated inflammation and cytokine production. S100A8 can also cause apoptosis by attracting CD8+ and natural killer (NK) cells to the skin [53]. The study also established the importance of oxidative stress in OLP; reactive oxygen species (ROS) cause keratinocyte apoptosis and dysfunction, and ROS can be generated in a vicious cycle by TCD4+ lymphocytes infiltrating OLP. In a case-control study involving 62 patients and 30 healthy people, oxidative stress in OLP was described and evaluated in 2016 [54]. The authors found substantial variations between patients and controls in total antioxidant potential (TAC, calculated using the Benzie and Strain method [36]), glutathione (GSH, assessed spectrophotometrically), and thiobarbituric-acid-reactive substances (TBARS, determined using the Aust method), which are lipid peroxidation products [55]. TAC and GSH levels were lower in OLP patients than in safety controls, as predicted, while TBARS levels were higher. Patients with an erosive type of lichen had higher values, suggesting extreme oxidative stress and a good correlation with clinical characteristics. These results may support the use of antioxidants in the mouth or on the skin [56]. A randomized, double-blind, parallel-group study was performed by Tvarijonaviciute et al. The sample consisted of 55 clinically and histopathologically diagnosed OLP patients. Twenty-six patients were treated with 2% Chamaemelum nobile gel and 29 with a placebo. Nonstimulated saliva was collected on the first day of the study and 4 weeks later. Salivary total antioxidant status (TAS) was evaluated by four different methods. The findings of the study indicated that changes in TAS in saliva are related to increased discomfort, xerostomia, and reduced drainage in patients with OLP, indicating that the patient’s health is declining. The use of Chamaemelum nobile gel for disease stabilization was recommended by the Authors [57].

Several studies have investigated the role of cortisol in oral lichen [58,59,60,61,62,63]. Cortisol is a biological marker of stress and anxiety, and its levels can change cytokine profiles [64]. Oral lichen has a dual relationship with stress: fear and traumatic experiences are thought to be a cause for OLP onset, but oral lichen itself is a source of stress for patients. The assessment of salivary cortisol in this complex situation seems to be synonymous with the ancestral debate, “Which came first, the chicken or the egg?” In reality, research results are inconsistent, and cortisol is unlikely to be useful as a biomarker in OLP. OLP is a T-cell-driven condition, as previously stated; however, it is still uncertain if the inflammation is caused by Th1 or Th2 expression. In particular, several cytokines are released in OLP by both recruited lymphocytes and damaged keratinocytes, resulting in a self-amplification process [65]. The measurement of specific interleukin in saliva is an excellent way to identify biomarkers in OLP and, more importantly, to develop targeted therapies. The findings for IL-6 and IL-8 are now more reliable. Interleukin-6 is active in B- and T-cell differentiation, and it has been shown to be able to inactivate p53 as cancer progresses [66]. In a meta-analysis, Mozaffari et al. found that IL-6 levels in saliva and serum of OLP patients were substantially higher than in stable controls, with saliva values being higher than serum values [67]. The group of Mozaffari conducted a meta-analysis on this topic [68]. The most intriguing discovery was that IL-8 [69] plays a crucial function in the transition of lichen from reticular to erosive, most likely due to keratinocyte restoring pathways losing functionality. The reduction in saliva following dexamethasone administration indicated the possible use of IL-8 in therapeutic monitoring [70].

TNF-α inhibitors dramatically changed the therapeutic scenario in dermatological inflammatory diseases [71,72]. Several studies have measured TNF-α levels in the saliva and serum of patients with OLP, reported different results. In fact, a high concentration of TNF-α plays a role in the progression of the pathologic events in OLP. The higher levels of TNF-α in saliva compared with those in serum suggest that measurement of this marker in saliva for treatment purposes may be more useful than measurements of this marker in serum. Despite its diagnostic value, saliva is a biological fluid affected by some confounding factors, such as age, type of OLP, stress, alcohol consumption, smoking and genetics; all those aspects must be taken into consideration while interpreting results and could be a limit [73]. Gavala et al. reported that alcohol can inhibit the levels of TNF-a in serum. Their results indicated that salivary TNF-a levels differed between clinical forms of OLP, with the erosive/atrophic form of the disease having the highest levels. This finding is quite predictable, as previously discussed the strong association between inflammatory cytokines’ levels and more severe forms of OLP [74]. In contrast, serum and salivary levels of interferon-γ (IFN-γ) are not related to clinical and severity appearance of OLP, as demonstrated by Mozaffari’s study group in a meta-analysis; probably, this cytokine has not a role in the pathogenesis of OLP [75]. In light of this review, future studies will have to confirm the role of TAS, IL-6 and IL-8 in OLP, especially in the neoplastic progression. In Table 1, the studies performed on the saliva of patients with psoriasis are summarized.

### 4.3. Blistering Diseases

Bullous pemphigoid (BP) and pemphigus vulgaris (PV) are two acquired bullous dermatoses that can affect the skin and/or mucous membranes. PV is an acantholytic bullous dermatosis characterized by the presence of blisters with a flaccid dome that appears on aflegmasic skin, expression of an intraepidermal disruption. It is common, especially in the early stages, the involvement of mucous membranes, in particular of the oral mucosa. BP is a non-acantholytic bullous dermatosis characterized by the appearance of sub-epidermal tense dome-shaped bullae. Unlike PV, the affected skin is highly flamed and itchy, and involvement of the oral mucosa is less frequent [76].

The diagnostic gold standard for the two diseases is a histologic examination with direct immunofluorescence of skin and/or mucosal biopsy. In PV the acantholysis and the intraepidermal detachment with characteristic intercellular IgG antibody deposition to desmoglein (Dsg) 1 and/or desmoglein 3, which are trans-membrane desmosomal proteins, are observed [77]. In bullous pemphigoid, on the other hand, histologically a sub-epidermal blister containing numerous eosinophils and neutrophils is observed and direct immunofluorescence shows linear deposits of C3 and IgG (BP180) at the level of the basement membrane [77].

In PV and less frequently in BP it is possible to demonstrate the presence of circulating antibodies in serum by indirect immunofluorescence techniques or by the most modern ELISA techniques [76]. Starting from this technique, some authors have proposed the use of saliva as a substrate for the research of BP180 and Dsg1 and 3. In 2006, Andreadis et al. demonstrated that ELISA analysis of Dsg3 and Dsg1 in the saliva is a highly sensitive and specific test that is suitable for diagnostic purposes, monitoring of disease activity and early detection of pemphigus relapses, as there is a high concordance between serum and salivary levels of the proteins [78]. The same was not found for pemphigoid as the BP180 ELISA kit, with recombinant non-collagen extracellular domain (NC16a), proved to be inadequate to detect circulating MMP antibodies in serum and saliva [78]. Similar results emerged from Ali’s study [79] on Dsg1 and 3, no further studies have been performed on BP. The potential of salivary testing in PV prognosis and mucosal severity has been investigated in two studies. De et al. included 43 patients with histologically confirmed PV and performed ELISA for Dsg1 and 3 on serum and saliva samples [80]. Although the authors demonstrated a significant correlation between serum and salivary tires for both Dsg1 and Dsg3 in PV, no correlation was found between oral mucosal Autoimmune Bullous Skin Disorder Intensity Score ABSIS and either serum or salivary Dsg1 and Dsg3 levels. Differently, Koopai et al. demonstrated a moderate significant correlation between Dsg1 and Dsg3 levels present on saliva detected by ELISA technique and PV severity, assessed by pemphigus disease area index (PDAI) score [81]. Similar results emerged from Mortazavi’s study [82]. In contrast to the previously discussed research, one Italian study was designed to assess the use of a BIOCHIP approach compared with ELISA in PV [83]. In fact, the authors considered saliva an unsuitable substrate for autoantibody detection because of the discordance between techniques found when using saliva samples. The data provide preliminary evidence to suggest that Dsg1 and Dsg3 detection in saliva can be a useful diagnostic and prognostic tool in patients with bullous diseases. In Table 2 the studies performed on the saliva of patients with blistering diseases are summarized.

### 4.4. Vitiligo

Vitiligo is an acquired skin disease of unknown origin characterized by the presence of achromic patches due to the loss of functioning melanocytes in the skin or hair, or both [84]. There are not many studies in the literature analyzing the content of saliva in patients with vitiligo. Sehgal et al. analyzed the saliva content of 76 vitiligo patients and evaluated the secretion of blood group-specific substances in saliva of vitiligo patients and normal controls, finding a statistically significantly greater distribution of secretors in the former as compared to the latter [85]. No further studies have been performed on this topic. In light of these findings, few conclusions can be drawn from the relevance of saliva collection in vitiligo. Table 3 summarizes the studies performed on the saliva of patients with vitiligo.

### 4.5. Atopic Dermatitis

Atopic dermatitis (AD) is a chronic inflammatory skin disease characterized by itch and eczematous lesions [86]. A multifactorial etiology leads to the clinical variety of AD combining genetic predisposing factors to environmental factors that together trigger a complex pathophysiological mechanism dictated by an imbalance in the response of T helper (Th) 1 and 2 lymphocytes with a predominance of the Th2 response. The diagnosis of AD is mainly clinical even if some laboratory parameters may be increased compared to healthy subjects. These differences have encouraged the search for biomarkers useful in the diagnosis, evaluation and prognosis of the disease: total serum IgE levels, eosinophilic cationic protein (ECP), IL-2R and thymus and activation-regulated chemokine (TARC/CCL17), have been detected as serum biomarkers for disease severity [87]. Nevertheless, none of these biomarkers alone is specific for disease severity due to the large number of biological pathways involved in the pathogenesis of AD and the clinical heterogeneity. Thijs JL et al. [87], in their communication published in 2015, sustained that a combination of biomarkers can overcome these problems and they also have proposed alternative ways than blood to measure biomarkers. Since the collection of blood is invasive and less suitable because of the need for trained personnel, Thijs and colleagues identified dried blood spots (DBS) and saliva as potential alternatives [87]. Since salivary glands are highly permeable and are surrounded by capillaries, saliva is like a mirror of circulating blood, with the advantage of non-invasive sampling. Nevertheless, the composition and protein concentrations in saliva are influenced by many factors such as age, sex, hydration status, flow rate, sampling and for this reason, a standardization of the methods for collection and handling of saliva samples is needed before introduction in daily practice. Two studies in 2013 have evaluated the salivary cortisol level in adults [78] and children [88] with AD. Mizawa et al. [89] considered salivary cortisol as a biomarker useful to assess the level of stress in adult AD patients. They enrolled 30 patients suffering from AD and 42 healthy subjects and they compared salivary cortisol levels between two groups. The salivary cortisol level in AD patients ranged from 0.47 to 5.18 ng/mL (1.97 ± 0.22 ng/mL; mean ± SE), which was significantly higher compared to that of the healthy controls (from 0.028 to 0.334 ng/mL; 0.11 ± 0.01 ng/mL; *p* < 0.01). The authors showed that the levels of salivary cortisol were significantly correlated with disease severity measured by the SCORAD index (*r* = 0.42, *p* < 0.05), but there was not a statistically significant correlation with the other serum biomarker levels (TARC, IgE, and LDH or the number of peripheral blood eosinophils) and skindex-16. High basal levels of cortisol suggest a hyporesponsiveness of the hypothalamic–pituitary–adrenal (HPA) axis to stress. Since cortisol is a potent attenuator of inflammatory reactions, poor responsiveness of the HPA axis under stress may be one explanation for stress-induced exacerbation of atopic dermatitis (AD) symptoms. Kojima et al. [88] studied the salivary cortisol response to stress in young children with AD. They enrolled 38 young patients with mild to severe AD according to the SCORAD index (mild < 25, moderate 25–50, severe > 50) and they measured the salivary cortisol level before and after a venipuncture, considering venipuncture as a stressor event. Contrary to the previous study, the authors noted that there were no significant differences in prevenipuncture cortisol levels between the groups with different AD severity and, in addition, the cortisol level in each group of subjects before venipuncture was compatible with the normal salivary cortisol level in healthy young children (approximately 0.16 to 0.36 lg/dL). In response to venipuncture as a stressor event, the salivary cortisol correlated negatively with the severity of AD. Finally, the authors found no significant correlation between salivary cortisol level and previous TCS treatment suggesting that the disease activity of AD, rather than TCS use, is responsible for dysfunction of the HPA axis in patients with severe AD [89]. In our opinion, future research will have to focus on the reliability of cortisol as a biomarker in saliva in patients with chronic inflammatory background status. Table 4 summarizes the characteristics of the two studies.

## 5. Conclusions

This review emphasizes the great potential of saliva proteomics analysis in a large number of heterogeneous skin inflammatory and immune-mediated diseases (skin IMID). Currently, not all molecular and pathophysiological mechanisms underlying most parts of skin IMID are established yet. It can be inferred that saliva could represent a useful and easy to obtain biological fluid in order to evaluate how inflammatory status can changes in the human organism in course of skin IMID.

Taking into account all reported data, we could postulate to draw a gradient of evidence on the relevance of available data on proteomics in skin IMID as follows: oral lichen planus, psoriasis, atopic dermatitis, blistering diseases, and vitiligo.

Although nowadays proteomic technologies are complex, and of limited accessibility, an explosion in -omics research applications in the next years is foreseen, following the future introduction of simple, and less expensive instruments, able to be applied to small salivary samples for early diagnosis of different systemic pathologies, as routine analysis.

## Figures and Tables

**Figure 1 ijms-22-07018-f001:**
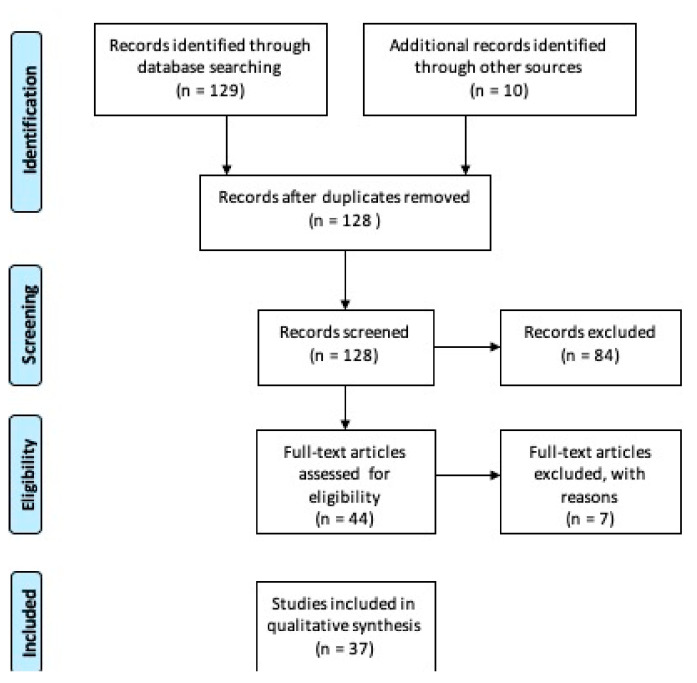
Preferred Reporting Items for Systematic Reviews and Meta-Analysis (PRISMA) on saliva proteomics data in immune mediated and inflammatory skin diseases. Research dates range from 1994 to 2020 [25].

**Table 1 ijms-22-07018-t001:** Summary of the studies performed on the saliva of patients with psoriasis.

First Author (Ref.), Year	Type of Molecules Studied in Saliva	Method Used for Analysis	Type of Study	Number of Patients	Results
H Fadel, 2013 [26]	Unstimulated salivary SR -Stimulated salivary SR -Salivary pH	-Unstimulated and paraffin-stimulated saliva samples collection, for the determination of secretion rate and buffer capacity	-Case-control study	-89 patients with mild-to-moderate psoriasis -54 individuals without psoriasis	Individuals with psoriasis had low salivary pH, compared to the control group.
F Asa’ad, 2018 [31]	2-IgA -CRP -sAA -Haptoglobin -K+ -TNFα -TGF-β1 -IL-1β -MCP-1	-Salivary level of IgA was assessed by radial immunodiffusion -Levels of salivary CRP and Haptoglobin were determined by an immunoturbidimetric method -sAA and K+ levels were analyzed using ISE (Ion Selective Electrode) technology for electrolyte measurements and LISA 500 plus systems for sAA -TNFα, TGF-β1, IL-1β, MCP-1 were assessed by using multianalyte ELISA Arrays	Review	-60 patients with psoriasis and 40 individuals without psoriasis, enrolled in the study concerning IgA, CRP, Haptoglobin. -20 patients with uncomplicated psoriasis and 20 individuals without psoriasis, enrolled in the study concerning sAA and K+ -60 patients with psoriasid and 45 patients without psoriasis, enrolled in the study concerning TNFα, TGF-β1, IL-1β, MCP-1	No statistically significant difference in the salivary level of IgA between psoriasis patients and healthy controls. Psoriasis patients with PASI > 10 had tendency to show lower levels of IgA, compared to patients with a PASI < 10. Patients with psoriasis had higher levels of CRP, Haptoglobin, sAA, K+, TNFα, TGF-β1, IL-1β, MCP-1, compared with controls.
R A Soudan, 2011 [32]	-K+ -Na+ -Cl− -sAA	-ISE (Ion Selective Electrode) technology for K+, Cl−, Na+ measurement. -LISA 500 plus system for sAA measurement.	-Case-control study	-20 patients with uncomplicated psoriasis -20 individuals without psoriasis	Psoriatics had significantly higher K+ and sAA concentrations than the controls, whereas there was no significant rise in the other salivary ions studied.
D Belstrom, 2020 [33]	-NGAL (neutrophil gelatinase-associated lipocalin)-Transferrin	Stimulated saliva samples were characterized by means of next-generation sequencing of the 16S rRNA gene. Salivary levels of NGAL and transferrin were quantified using immunoassays.	-Case-control study	-27 patients with psoriasis -58 patients with periodontitis -52 orally healthy individuals	Significantly lower mean salivary levels of NGAL and transferrin were identified in patients with psoriasis, compared to patients with periodontitis and orally healthy controls.
G Ganzetti, 2016 [34]	-IL-1β	-IL-1β levels were evaluated via an enzyme-linked immunosorbent assay kit	-Case-control study	-25 patients with psoriasis -20 individuals without psoriasis	At baseline, patients had significantly higher salivary IL1β levels than controls. In patients with psoriasis, TNF-a inhibitor treatment resulted in significantly reduced IL1β levels compared with baseline, but IL1β levels remained significantly higher than in control subjects even after treatment.
G Ganzetti, 2015 [35]	-IL-1β -IL-6-TGF -β1- IL-8 -TNF-β -IFN-χ -IL-17A -IL-4 -IL-10 -MCP-1 -MIP-1α,β	Multi-Analyte ELISA array Kit	-Case-control study	-60 patients with psoriasis -45 individuals without psoriasis	Patients with active psoriasis had significantly higher salivary IL1β, TNF-α, TGF-β, and MCP-1 levels than healthy controls.
A Skutnik-Radziszewska, 2020 [38]	-TNF-α -IL-2-IFN -χ -IL-10	ELISA	-Case-control study	-30 patients with psoriasis and hyposalivation -30 patients with psoriasis and normal secretion of saliva -60 individuals without psoriasis	The levels of tumor necrosis factor-alpha (TNF-α), interleukin-2 (IL-2), and interferon-gamma (INF-γ) were significantly higher, whereas interleukin-10 (IL-10) content was considerably lower in unstimulated and stimulated saliva of patients with psoriasis compared to the controls.
A Skutnik-Radziszewska, 2020 [39]	-Peroxidase (Px) -Catalase (Cat) -Superoxide Dismutase (SOD)	The activity of antioxidant enzymes (Px, CAT, and SOD) was measured in NWS (unstimulated saliva), SWS (stimulated saliva), and erythro- cytes by performing Redox Analysis. Absorbance/fluorescence was measured with an Infinite M200 PRO Multimode Tecan microplate reader.	-Case-control study	-40 patients with psoriasis -40 individuals without psoriasis	The concentration of Px, CAT, and SOD was significantly higher in NWS of patients with plaque psoriasis vs. healthy subjects. In SWS of psoriatic patients, there was a considerably higher concentration of Px and CAT.
A Bahramian, 2018 [40]	-Vitamine D	Vitamin D total (25-hydroxy vitamin D) kit was used with the electrochemiluminescence technique to determine and compare salivary and serum levels of vitamin D between the healthy individuals and those with RAS.	-Case-control study	-26 patients with RAS (Recurrent aphtous stomatitis) -26 healthy individuals	The serum levels of vitamin D in patients with RAS were significantly less than that in healthy individuals; however, there were no significant differences in salivary vitamin D levels between patients with RAS and healthy individuals. There was a significant and positive correlation between serum and salivary levels of vitamin D in all patients.
U Bottoni, 2016 [41]	-Saliva proteomic components	It was performed attenuated total reflection (ATR) in conjunction with infrared spectroscopy.	-Case-control study	-35 patients with psoriasis -20 patients with diabetes -20 healthy individuals	There were differences in the secondary structure composition of proteins between psoriatic and diabetic patients as compared to the control group. Saliva spectra of the control group and that of the palmoplantar psoriatic patients differ from plaque psoriasis and diabetic patient spectra because of the absence of the amide II band and the presence of different secondary protein-structure conformations.
Y Li, 2020 [42]	-Differential expressed proteins (DEPs)	Tandem mass tags (TMTs) coupled with liquid chromatography–mass spectrometry (LC–MS)/MS.	-Case-control study	-11 patients with psoriasis -11 individuals without psoriasis	A total of 4562 differentially expressed proteins (DEPs) between PV lesional tissues and healthy tissues were identified.

**Table 2 ijms-22-07018-t002:** Summary of the studies performed on the saliva of patients with blistering diseases.

First Author [Ref.], Year	Type of Molecules Studied in Saliva	Method Used for Analysis	Type of Study	Number of Patients	Results
Andreadis, 2006 [78]	Anti-Desmoglein 1 and desmoglein 3 antibodies in PV; -Anti-BP180 antibodies in BP.	ELISA	Case-control study (?)	-12 patients with MMP; -12 patients with BP; -10 patients with PV; -10 healthy controls.	ELISA analysis of Dsg3 and Dsg1 in saliva is a highly sensitive and specific test that is suitable for diagnostic purposes, monitoring of disease activity and early detection of pemphigus relapses, as there is a high concordance between serum and salivary levels of the proteins.
Ali, 2016 [79]	Anti-Dsg3 IgA antibodies in PV.	ELISA	Case-control study (?)	-23 patients with PV -17 healthy subjects -9 disease controls	Assay of salivary IgG antibodies to Dsg3 offers a diagnostic alternative to serum in the diagnosis and monitoring of PV. The role of anti-Dsg3 IgA antibodies requires further elucidation in the pathogenesis of PV.
De, 2017 [80]	-Anti-Dsg1 and 3 antibodies in PV.	ELISA	Case-control study (?)	-43 patients with PV; -5 controls	There was a statistically significant correlation between serum and salivary Dsg1 levels and between serum and salivary Dsg3 levels. There was no correlation between serum or salivary Dsg1 and Dsg3 levels with the objective component of the oral mucosal Autoimmune Bullous Skin Disorder Intensity Score (ABSIS).
Koopai, 2018 [81]	-Anti-Dsg1 and anti-Dsg3 antibodies in PV.	ELISA	Cross-sectional study	-50 patients with PV	Moderate significant correlation between Dsg1 and Dsg3 levels present on saliva detected by ELISA technique and PV severity.
Mortazavi, 2015 [82]	-Anti-Dsg1 and anti-Dsg3 antibodies in PV.	ELISA	Case-control study	-86 untreated PV; -80 age- and sex-matched PV-free controls.	Salivary anti-Dsg 1 and 3 ELISA with high specificities (98.9%) could be suggested as safe and noninvasive methods for the diagnosis of PV when obtaining a blood sample is difficult.
Russo, 2017 [83]	-Anti-Dsg1 and anti-Dsg3 antibodies in PV.	ELISA and BIOCHIP Approach	Pilot study	-8 patients with PV	Autoantibodies to DSG3 were detected in 8 out of 8 salivary samples by ELISA and in 6 out of 8 salivary samples by the BIOCHIP approach. Autoantibodies to DSG1 were negative in all salivary samples using both ELISA and BIOCHIP. There were no positive results in the negative control group. In conclusion, the results of this pilot study indicate a lack of correlation between serum and salivary results using both ELISA and BIOCHIP, indicating that saliva may not be the ideal substrate for the laboratory diagnosis of PV using these approaches.

**Table 3 ijms-22-07018-t003:** Summary of the studies performed on the saliva of patients with vitiligo.

First Author [Ref.], year	Type of Molecules Studied in Saliva	Method Used for Analysis	Type of Study	Number of Patients	Results
Sehgal, 1967 [85]	Not declared.	Not declared.	-Case-control study	-76 patients with vitiligo. -370 normal controls	The study revealed an increased predilection for the secretors to develop the disease as compared to non-secretors. More investigations in this connection are indicated which may further enrich our information as regards the transmission and course of the disease

**Table 4 ijms-22-07018-t004:** Summary of the studies performed on the saliva of patients with atopic dermatitis.

First Author [Ref.], Year	Demography	Time and Duration of Collection	Storage of Sample	Kind of Sample	Salivary Analysis
Mizawa M, 2013 [88]	30 adults (15 males and 15 females; age, 15–62 years; mean age, 29.6 years) SCORAD index (mean ± SE) ranged from 9.9 to 80.3 (46.7 ± 3.2)	9–11 a.m. 5 min	Centrifugation Supernatant stored at −80 °C	Twisted cotton dental cord (Salimetrics LLC, State College, PA, USA)	linked immunosorbent assay kits (1-3002; Salimetrics LLC, State College, PA, USA) plate reader (450 nm measurement wavelength; ARVO MX; Perkin Elmer Life Science, Boston, MA, USA)
Kojima R, 2013 [89]	38 young children (24 boys and 14 girls) median age 16.5 months, range 3–66 months) SCORAD INDEX median (range) mild [n12] 16 (8–25) moderate [n14] 40 (26–48) severe [n12] 64.5 (51–86)	10 a.m.–3 p.m. 5 min before venipuncture 15–20 min after venipuncture 1 min under the tongue	Centrifugation 15 min at 1800× *g* Supernatant stored at −30 °C	Sorbette sampling device (Salimetrics, State College, PA, USA)	salivary cortisol enzyme-linked immunosorbent assay kit (Salimetrics), according to the manufacturer’s protocol.

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
