# Peer review of "Saliva Proteomics as Fluid Signature of Inflammatory and Immune-Mediated Skin Diseases"

_ijms, 2021, doi:10.3390/ijms22137018_

Round 1

Reviewer 1 Report

Title: Saliva proteomics as fluid signature of inflammatory and im-mune mediated skin diseases.

The authors make a review on proteins isolated in saliva, related to diagnostic and prognostic factors of five dermal pathologies. An issue of self-citation is also detected in the review.

The following recommendations could improve the review:

-The authors should include in the introduction a description of the techniques of large-scale experimental analysis of proteins and proteomes, define a little what proteomics consists of.

-In the Identify relevant studies section, other electronic databases such as Cochrane Library, Embase, Lilacs, Cinahl and gray literature databases are missing. Do not the authors believe that there are many bases without consulting that could not include important information?.

-They consult Pubmed, but all key words are not MeSH terms. This can also represent an important bias in the information obtained.

-In this same section, the authors should specify in the inclusion criteria, what type of studies, observational, clinical trials, in vitro ...... have been included in the review.

-In the discussion section, it would be very clear that at the end of each pathology, it was concluded with the most significant proteins of a diagnostic-prognostic character, from the point of view of the authors.

-The authors self-cite, up to 10 times. This represents more than 10% of the citations in the manuscript. They believe that all their self-citations are relevant or contribute information to the manuscript. Self-cite abuse is also considered unethical.

Author Response

The authors make a review on proteins isolated in saliva, related to diagnostic and prognostic factors of five dermal pathologies. An issue of self-citation is also detected in the review.

The following recommendations could improve the review:

-The authors should include in the introduction a description of the techniques of large-scale experimental analysis of proteins and proteomes, define a little what proteomics consists of.

RESPONSE: A brief description of techniques of experimental analysis of proteins, in order to better define what proteomics consist of, has been included.

-In the Identify relevant studies section, other electronic databases such as Cochrane Library, Embase, Lilacs, Cinahl and gray literature databases are missing. Do not the authors believe that there are many bases without consulting that could not include important information?

RESPONSE: Consultation of other databases eg. Embase had not brought out any further publications on the topic of the review

They consult Pubmed, but all key words are not MeSH terms. This can also represent an important bias in the information obtained.

RESPONSE: It is well known that untagged terms that are entered in the PubMed search box are automatically mapped to the MeSH vocabulary when a match is found. However we have chosen to search a MeSH heading and a subheading combination using the AND Boolean operator.

-In this same section, the authors should specify in the inclusion criteria, what type of studies, observational, clinical trials, in vitro ...... have been included in the review.

RESPONSE: in the section inclusion criteria have been changed as follows: "studies reporting on saliva proteomics in IMID, studies published in the English language, abstract available, no restriction on design of the study was considered, and randomized controlled trial, case-control study, cross-sectional study, case reports and series."

 -In the discussion section, it would be very clear that at the end of each pathology, it was concluded with the most significant proteins of a diagnostic-prognostic character, from the point of view of the authors.

RESPONSE: At the end of each pathology, a statement including the most significant proteins with diagnostic/prognostic relevance has been included.

 -The authors self-cite, up to 10 times. This represents more than 10% of the citations in the manuscript. They believe that all their self-citations are relevant or contribute information to the manuscript. Self-cite abuse is also considered unethical.

RESPONSE: As authors reported self-citations for 10 articles from 90 articles reported in the references list, it cannot be considered a self-cite abuse. Moreover, to cite previous experience in the field of immunomediated skin diseases could reinforce the translational message of the review.

Reviewer 2 Report

A problem which is reviewed in the paper is a saliva proteomics to find biomarkers or their signatures for skin disease. Diagnostics based on one or more proteins is widely proposed in those diseases that start cryptically without early symptoms able to be identified by a clinician. Skin diseases manifest themselves at the early stage and it is not clear why we need to study saliva proteome with a labor-consuming and equipment-sensitive technique, such as proteomics. Moreover, in contract to plasma and other internal and homeostatic body fluids, saliva is excreted and its proteome is in more extent variable due to various environmental and dietary factors, which makes it difficult to get significant biomarkers. A difference between saliva and plasma is illustrated by a ratio between paper counts in Pubmed, where it is ca. 1:9 between saliva and plasma proteomics, despite higher technical availability of the former fluid. I do not think, thus, that a suggested review based on the limited number of papers with a big variation in methods used for proteomics attracts interests of a broad readership of the Journal.           

Author Response

Many thanks for your review and comments.

Below are the reasons that led us to conduct the systematic review on this topic.

  1. We agree that published papers on saliva are limited, but we believe that a systematic review of existing papers can provide an opportunity to explore the need for research on this biological fluid as opposed to blood.
  2. Although we agree that diagnostics based on one or more proteins is appropriate for those diseases that start cryptically without early symptoms identifiable by a clinician (while skin diseases manifest themselves at the early stage). However, skin disease usually show overlap clinical signs which make diagnosis more complicated then expected. Moreover, skin inflammatory and immune mediated diseases usually show a wide range of therapeutic response to commonly avalilable treatments. For all these reasons, proteomics seems to be promising both for increasing the early diagnosis of skin and mucous diseases, and forpredicting therapeutic response to available treatments
  3. Although we agree that the salivary proteome may be more influenced by environmental and dietary factors (compared to plasma proteome), we believe that the low invasiveness of the method to obtain salivary biological fluid explains the need to implement research in this field.

Round 2

Reviewer 1 Report

The manuscript has been improved in some of the 6 recommendations that this reviewer made. There is still one issue that should be modified. In key words, there are two words, Oral lichen plant, and blistering diseases, that should be modified. Because if the manuscript was approved, they would avoid appearing appropriately in the bilbiographical repertoires. The modifications made in the introduction should appear in another color to facilitate the reviewer's work. The first author of the review, A. Campanati, appears self-cited up to 13 times, most of the other authors, about 10 times. It is a clear example of self-citation abuse. If proteins in saliva are reviewed in 5 skin diseases, an article on a novel therapeutic approach in a disease should not be cited. Please include only relevant references. 

Author Response

Dear Editor,

the manuscript has been revised according the reviewer's suggestion, and we hope this version will be suitable for publication on JMolSci.

Herein are reported our responses to reviewer 1 point by point:

The manuscript has been improved in some of the 6 recommendations that this reviewer made.There is still one issue that should be modified. In key words, there are two words, Oral lichen plant, and blistering diseases, that should be modified. THEY HAVE BEEN MODIFIED

The modifications made in the introduction should appear in another color to facilitate the reviewer's work. DONE

The first author of the review, A. Campanati, appears self-cited up to 13 times, most of the other authors, about 10 times. It is a clear example of self-citation abuse. If proteins in saliva are reviewed in 5 skin diseases, an article on a novel therapeutic approach in a disease should not be cited.Please include only relevant references. ONLY RELEVANT REFERENCES HAVE BEEN INCLUDED, MOREOVER THE REFERENCES ON NOVEL THERAPEUTIC APPROACHES HAS BEEN REMOVED.